# Cortical tracking of speech in noise accounts for reading strategies in children

**Florian Destoky**[1]*, **Julie Bertels**[1,2], **Maxime Niesen**[1,3], **Vincent Wens**[1,4], **Marc Vander Ghinst**[1], **Jacqueline Leybaert**[5], **Marie Lallier**[6], **Robin A. A. Ince**[7], **Joachim Gross**[7,8], **Xavier De Tiège**[1,4], **Mathieu Bourguignon**[1,5,6]

**1** Laboratoire de Cartographie fonctionnelle du Cerveau, UNI–ULB Neuroscience Institute, Université libre de Bruxelles (ULB), Brussels, Belgium, **2** Consciousness, Cognition and Computation group, UNI–ULB Neuroscience Institute, Université libre de Bruxelles (ULB), Brussels, Belgium, **3** Service d'ORL et de chirurgie cervico-faciale, ULB-Hôpital Erasme, Université libre de Bruxelles (ULB), Brussels, Belgium, **4** Department of Functional Neuroimaging, Service of Nuclear Medicine, CUB Hôpital Erasme, Université libre de Bruxelles (ULB), Brussels, Belgium, **5** Laboratoire Cognition Langage et Développement, UNI–ULB Neuroscience Institute, Université libre de Bruxelles (ULB), Brussels, Belgium, **6** BCBL, Basque Center on Cognition, Brain and Language, San Sebastian, Spain, **7** Institute of Neuroscience and Psychology, University of Glasgow, Glasgow, United Kingdom, **8** Institute for Biomagnetism and Biosignal analysis, University of Muenster, Muenster, Germany

☯ These authors contributed equally to this work.
* florian.destoky@ulb.ac.be

**Data Availability Statement:** The data and the code that support the findings of this study are available on the Open Science Framework at "https://osf.io/9ce5t/". The underlying numerical

## Abstract

Humans' propensity to acquire literacy relates to several factors, including the ability to understand speech in noise (SiN). Still, the nature of the relation between reading and SiN perception abilities remains poorly understood. Here, we dissect the interplay between (1) reading abilities, (2) classical behavioral predictors of reading (phonological awareness, phonological memory, and rapid automatized naming), and (3) electrophysiological markers of SiN perception in 99 elementary school children (26 with dyslexia). We demonstrate that, in typical readers, cortical representation of the phrasal content of SiN relates to the degree of development of the lexical (but not sublexical) reading strategy. In contrast, classical behavioral predictors of reading abilities and the ability to benefit from visual speech to represent the syllabic content of SiN account for global reading performance (i.e., speed and accuracy of lexical and sublexical reading). In individuals with dyslexia, we found preserved integration of visual speech information to optimize processing of syntactic information but not to sustain acoustic/phonemic processing. Finally, within children with dyslexia, measures of cortical representation of the phrasal content of SiN were negatively related to reading speed and positively related to the compromise between reading precision and reading speed, potentially owing to compensatory attentional mechanisms. These results clarify the nature of the relation between SiN perception and reading abilities in typical child readers and children with dyslexia and identify novel electrophysiological markers of emergent literacy.

data for each figure can also be found in the supporting data files.

**Funding:** F.D., J.B. and M.B. were supported by the program Attract of Innoviris (https://innoviris. brussels/; grant number 2015-BB2B-10). J.B. was supported by a research grant from the Fonds de Soutien Marguerite-Marie Delacroix (https://www. fondsmmdelacroix.org/). R.A.A.I. was supported by the Wellcome Trust (https://wellcome.ac.uk/; grant number 214120/Z/18/Z). X.D.T. was Postdoctorate Clinical Master Specialist at the Fonds de la Recherche Scientifique (F.R.S.-FNRS, https:// www.frs-fnrs.be/en/). M.B. was supported by the Spanish Ministry of Economy and Competitiveness (https://www.ciencia.gob.es/; grant number PSI2016-77175-P), and by the Marie Skłodowska-Curie Action of the European Commission (https:// ec.europa.eu/research/mariecurieactions/msca-actions_en; grant number 743562). This study and the MEG project at the CUB Hôpital Erasme are financially supported by the Fonds Erasme (https:// www.fondserasme.org/fondserasme_en.html; Research Convention "Les Voies du Savoir"). The funders had no role in study design, data collection and analysis, decision to publish, or preparation of the manuscript.

**Competing interests:** The authors have declared that no competing interests exist.

**Abbreviations:** CTS, cortical tracking of speech; IQ, intelligence quotient; MEG, magnetoencephalography; nCTS, normalized CTS; PID, partial information decomposition; RAN, rapid automatized naming; SiN, speech in noise; TAP, test of attentional performance.

## Introduction

Acquiring literacy is tremendously important in our societies. Central for reading acquisition are adequate phonological awareness [1–3], phonological memory [4,5], and rapid automatized naming (RAN) [6–8]. The adequacy of the learning environment also plays a major role [9,10]. In particular, the presence of recurrent noise in the learning environment can substantially hinder reading acquisition [11,12]. Therefore, the ability to understand speech in noise (SiN)—which is known to differ among individuals [13,14]—should modulate the negative impact of environmental noise on reading acquisition. And indeed, the quality of brainstem responses to syllables in noise predicts reading abilities and its precursors [15]. Moreover, individuals with dyslexia often exhibit a SiN perception deficit [16,17] that is particularly apparent when the background noise is composed of speech [18]. This deficit has been hypothesized to be rooted in a deficit in phonological awareness [19,20], but contradictory reports do exist [21]. The question of whether SiN processing abilities relate to reading because of a common dependence on classical behavioral predictors (i.e., phonological awareness, phonological memory, and RAN) or other cognitive or neurophysiological processes specific to SiN processing is thus open. Furthermore, which aspects of reading and SiN processing abilities are related is also unexplored. Understanding these relations is especially important given that acoustic noise is ubiquitous and given how adverse dyslexia can be for the cognitive and social development of children.

Reading is a multifaceted process. Hence, it is reasonable to think that SiN processing might relate to a restricted set of aspects of reading. Following the dual-route cascaded model, reading in languages with alphabetic orthographies is supported by two separate routes: the sublexical and the lexical routes [22,23], which do interact following other models of reading [24]. The sublexical route implements the grapheme-to-phoneme conversion. It is used when reading unfamiliar words or pseudowords, but it is not suitable for correctly reading irregular words (i.e., yacht) and acquiring fluent reading. Skilled reading relies on the lexical route, which supports fast recognition of the orthographic word form of familiar words. The lexical route is indispensable for reading irregular words, and it benefits the reading of regular words much more than the reading of pseudowords. Remarkably, the brain would implement these two reading strategies in two distinct neural pathways, mostly in the left hemisphere [25–29].

There are also several distinct aspects of SiN processing that could relate to reading, and these can be derived from electrophysiological recordings of brain activity during connected-speech listening. When listening to connected speech, human auditory cortical activity tracks the fluctuations of speech temporal envelope at frequencies matching the speech hierarchical linguistic structures, i.e., phrases/sentences (0.2–1.5 Hz) and words/syllables (2–8 Hz) [30–40]. Such cortical tracking of speech (CTS) is thought to be essential for speech comprehension [33,35,37,39,41–43]. Most convincingly, speech intelligibility can be enhanced by speech-matched transcranial electrical stimulation of auditory cortices [42,44]. Corresponding brain oscillations would subserve the segmentation or parsing of incoming connected speech to promote speech recognition [33,34,39,41,45]. In SiN conditions, child and adult brains preferentially track the attended speech rather than the global auditory scene, though with reduced fidelity (especially reduced in the right hemisphere) when the noise hinders comprehension [30,31,40,46–56]. Assessing CTS in noise can therefore provide objective measures of the impact of noise on the cortical representation of the different hierarchical linguistic structures of speech. Also relevant is how SiN perception is impacted by noise properties. In essence, the relevant parameters for an acoustic noise in SiN conditions are the degree of energetic and informational masking [57]. The noise is energetic when it overlaps spectrotemporally with speech signal and is nonenergetic otherwise. The noise is informational when it is made up of

other speech signals (as in the case of a multitalker babble, even in an unknown language, but not time-reversed) and noninformational otherwise [58–60]. An energetic noise introduces physical interferences, and an informational noise introduces perceptual interferences. Finally, to enhance SiN processing, humans also benefit from visual information of the speaker's articulatory mouth movements [61,62]. All these aspects of SiN perception can be captured by measures of CTS.

In this study, we investigated the relations between reading abilities, neural representations of SiN quantified with CTS, and classical behavioral predictors of reading in elementary school children. To fully characterize cortical SiN processing, we measured CTS in several types of background noises introducing different levels of energetic and informational masking and in conditions in which the face of the speaker was visible ("lips") or not ("pics") while talking. This study was designed to answer four major questions: (1) What aspects of cortical SiN processing and reading abilities are related in typically developing elementary school children? (2) To what extent are these relations mediated by classical behavioral predictors of reading? (3) Are these different aspects of cortical SiN processing altered in children with dyslexia in comparison with typical readers matched for age or reading level? (4) What aspects of cortical SiN processing and reading abilities are related in children with dyslexia? As preliminary steps to tackle these questions, we identify relevant features of CTS in noise and assess in a global analysis the nature of the information about reading brought by all the identified features of CTS in noise and classical behavioral predictors of reading abilities.

## Results

We first report on 73 children with typical reading abilities. Then, we report on 26 children with dyslexia matched with a subsample of the 73 typical readers for age ($n$ = 26) or reading level ($n$ = 26). Both control groups were included to tell whether development or reading experience can explain potentially uncovered SiN deficits [63]. Reading performance and its classical behavioral predictors were characterized in a comprehensive cognitive evaluation (Table 1). Children's brain activity was recorded with magnetoencephalography (MEG) while they were attending to four videos of approximately 6 min each. Each video featured nine conditions: one noiseless and eight SiN resulting from the combination of four types of noise with lips or pics visual inputs (Fig 1, S1 Fig, and S1 Video). The opposite- and same-gender babble noises introduced informational interferences and a similar degree of energetic masking (see S1 Methods). The least- and most-energetic nonspeech noises introduced a degree of energetic masking in accordance with their naming but no informational interference.

For each condition, we regressed the temporal envelope of the attended speech on MEG signals with several time lags using ridge regression and cross validation (see Methods for details) [64]. The ensuing regression model was used to reconstruct speech temporal envelope from the recorded MEG signal. CTS was computed as the correlation between the genuine and reconstructed speech temporal envelopes. We did this for MEG and speech envelope signals filtered at 0.2–1.5 Hz (phrasal rate) [30,65] and 2–8 Hz (syllabic rate) [50,54,66,67] and for MEG sensor signals in the left and right hemispheres separately because the cortical bases of reading and SiN processing are hemispherically asymmetric [25–29,31,40].

S1 Table presents the percentage of the 73 typical readers showing statistically significant phrasal and syllabic CTS for both hemispheres and each condition. All typical readers showed significant phrasal CTS in noiseless and nonspeech noise conditions, and still most of them in babble noise conditions (mean ± SD across conditions, 98.3% ± 2.1%). Most of the typical readers showed significant syllabic CTS in noiseless and nonspeech noise conditions (93.8% ±

**Table 1. Mean and standard deviation of behavioral scores in each reading group of 26 children and comparisons (*t* tests) between groups.**

| Behavioral measure | Children with dyslexia | | Age-matched controls | | Reading level controls | | Children with dyslexia compared with controls | | | |
| --- | --- | --- | --- | --- | --- | --- | --- | --- | --- | --- |
| | | | | | | | in age | | in reading level | |
| | Mean | SD | Mean | SD | Mean | SD | *p* | t(df) | *p* | t(df) |
| Chronological age | 10.2 | 1.08 | 9.97 | 1.01 | 7.76 | 0.60 | 0.36 | 0.93 | **<0.0001** | **10.3** |
| Nonverbal IQ | 111 | 11 | 114 | 10 | 112 | 9 | 0.30 | −1.04 | 0.784 | −0.28 |
| Socioeconomic status | 6.12 | 2.44 | 6.96 | 1.45 | 6.96 | 2.47 | 0.14 | −1.50 | 0.17 | −1.40 |
| Alouette reading accuracy | 89.0 | 5.7 | 96.2 | 2.1 | 89.0 | 6.46 | **<0.0001** | **−6.07** | 0.988 | 0.01 |
| Alouette reading speed | 141 | 61 | 292 | 91 | 138 | 64 | **<0.0001** | **−7.04** | 0.867 | 0.17 |
| Irregular words reading (words/s) | 0.54 | 0.33 | 1.16 | 0.44 | 0.40 | 0.35 | **<0.0001** | **−5.82** | 0.15 | 1.47 |
| Regular words reading (words/s) | 0.73 | 0.41 | 1.35 | 0.41 | 0.61 | 0.35 | **<0.0001** | **−5.51** | 0.29 | 1.06 |
| Pseudowords reading (words/s) | 0.42 | 0.24 | 0.78 | 0.30 | 0.39 | 0.21 | **<0.0001** | **−4.88** | 0.61 | 0.50 |
| Visual attention | 30.3 | 3.74 | 32.0 | 2.69 | 27.4 | 4.43 | 0.070 | −0.95 | **0.014** | **2.53** |
| Phoneme suppression | 7.92 | 2.15 | 9.04 | 1.75 | 8.42 | 1.27 | **0.046** | **−2.05** | 0.313 | −1.02 |
| Phoneme fusion | 7.73 | 1.59 | 9.31 | 0.97 | 8.92 | 1.16 | **<0.0001** | **−4.32** | **0.003** | **−3.09** |
| Forward digit span | 5.08 | 0.84 | 5.8 | 0.69 | 5.15 | 0.78 | **0.001** | **−3.41** | 0.735 | −0.34 |
| Backward digit span | 3.69 | 0.79 | 4.5 | 1.33 | 3.38 | 0.75 | **0.011** | **−2.66** | 0.156 | 1.44 |
| RAN time (s) | 24.4 | 7.84 | 20.1 | 3.02 | 30.6 | 7.51 | **0.013** | **2.59** | **0.005** | **−2.91** |
| TAP mean response time (ms) | 627 | 99.0 | 613 | 75.4 | 667 | 93.4 | 0.59 | 0.53 | **0.07** | **−1.86** |
| TAP SD response time (ms) | 140 | 45.0 | 129 | 30.3 | 171 | 46.7 | 0.33 | 0.98 | **0.02** | **−2.36** |
| TAP correct responses | 15.6 | 0.58 | 15.7 | 0.68 | 15.3 | 1.07 | 0.42 | −0.81 | 0.11 | 1.65 |
| TAP false responses | 2.15 | 2.26 | 0.84 | 1.28 | 1.21 | 0.97 | **0.014** | **2.54** | 0.89 | 0.13 |

The number of df was 50 for all comparisons, except those involving auditory attention (TAP) scores (children with dyslexia versus controls in age, df = 49; children with dyslexia versus controls in reading level, df = 38) and socioeconomic status (children with dyslexia versus controls in age, df = 49; children with dyslexia versus controls in reading level, df = 47).

Abbreviations: df, degrees of freedom; IQ, intelligence quotient; RAN, rapid automatized naming; SD, standard deviation; TAP, test of attentional performance

3.2%) and slightly less of them in babble noise conditions (80.1% ± 4.3%). This result clearly indicates that CTS can be robustly assessed at the subject level.

S1 Data provides all participants' behavioral and CTS values on which the remainder of the results is based.

## What aspects of SiN processing modulate the measures of CTS in noise?

First, we identify the main factors modulating CTS in SiN conditions. To that aim, we evaluated with linear mixed-effects modeling how the normalized CTS (nCTS) in SiN conditions depends on hemisphere, noise properties, and visibility of the talker's lips. The nCTS is a contrast between CTS in SiN ($CTS_{SiN}$) and noiseless ($CTS_{noiseless}$) conditions defined as

$$nCTS = (CTS_{SiN} - CTS_{noiseless})/(CTS_{SiN} + CTS_{noiseless})$$

(see Methods for further technical details). It takes values between −1 and 1, with negative values indicating that the noise reduces CTS. Such contrast presents the advantage of being specific to SiN processing abilities by factoring out the global level of CTS in the noiseless condition. In that analysis, nCTS values were corrected (linear regression intertwined with outlier fixing) for age, time spent at school, and intelligence quotient (IQ) (see S2 Methods).

Table 2 presents the final linear mixed-effects model of phrasal and syllabic nCTS, and Fig 2 illustrates underlying values.

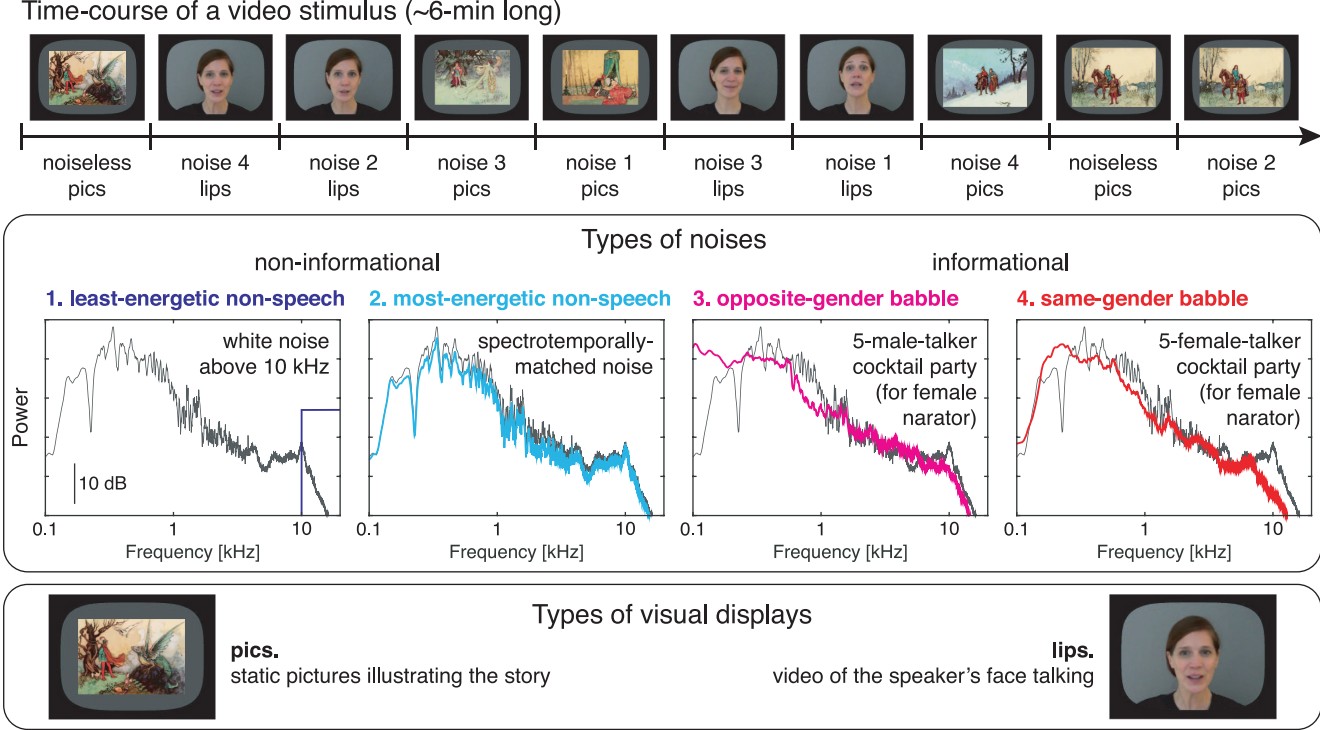

**Fig 1. Illustration of the experimental material used in the neuroimaging assessment.** Subjects viewed four videos of approximately 6 min in duration in which a different narrator (two females, two males) told a story. Each video was divided into 10 blocks to which experimental conditions were assigned. There were two blocks of the noiseless condition, and eight blocks of speech-in-noise conditions: one block for each possible combination of the four types of noise and two types of visual display. The interference introduced by the noise was either informational or not and varied in terms of degree of energetic masking. Power spectra are presented for all types of noise (colored traces) and one of the attended speeches (gray traces; here, that of a female narrator). The visual display provided visual speech information (lips) or not (pics).

The pattern of how nCTS changed with different types of noise was overall similar for phrasal and syllabic nCTS. Nonspeech noise did not substantially change CTS (nCTS was close to 0). However, babble noise resulted in a substantial reduction of CTS compared with

**Table 2. Factors included in the final linear mixed-effects model fit to the nCTS (independent variable) at phrasal rate and at syllabic rate.** Factors are listed in their order of inclusion.

| Factors | $\chi^2$ | | $p$ |
|---|---|---|---|
| | df | value | |
| Phrasal nCTS | | | |
| Noise | 3 | 598 | <0.0001 |
| Visual | 1 | 127 | <0.0001 |
| Hemisphere | 1 | 17.3 | <0.0001 |
| Noise × visual | 3 | 67.7 | <0.0001 |
| Noise × hemisphere | 3 | 11.0 | 0.012 |
| Syllabic nCTS | | | |
| Noise | 3 | 333 | <0.0001 |
| Visual | 1 | 21.1 | <0.0001 |
| Hemisphere | 1 | 10.5 | 0.0012 |

Abbreviation: nCTS, normalized cortical tracking of speech

## A. Phrasal nCTS     B. Syllabic nCTS

**Fig 2.** Impact of the main fixed effects on the nCTS at phrasal (A) and syllabic rates (B). Mean and SEM values are displayed as a function of noise properties. The four traces correspond to visual conditions with the speaker's talking face visible (lips; black traces) and with static pictures illustrating the story (pics; gray traces), within the left (lh; connected traces) and right (rh; dashed traces) hemispheres. nCTS values are bounded between −1 and 1, with values below 0 indicating lower CTS in speech-in-noise conditions than in noiseless conditions. S2 Data contains the underlying data for this figure. lh, left hemisphere; nCTS, normalized cortical tracking of speech; rh, right hemisphere.

the noiseless condition for both hemispheres and irrespective of the availability of visual speech information. That is, nCTS in babble noise conditions was roughly between −0.1 and −0.3, indicating that CTS in babble noise was 20%–50% (values obtained by inverting the formula of nCTS) lower than CTS in noiseless conditions.

Availability of visual speech information (lips conditions) increased the level of nCTS only in babble noise conditions for phrasal nCTS and in all noise conditions for syllabic nCTS.

And finally, the noise impacted nCTS differently in the left and right hemispheres. The phrasal nCTS was higher in the left than right hemisphere in babble noise conditions. It was the other way around for syllabic nCTS in all noise conditions.

Note that in the lips conditions, wherein participants saw the narrator's talking face, visual cortical activity driven by articulatory mouth movements could have contributed to nCTS values. However, such visual contribution was actually negligible (see S1 Results).

In summary, the CTS is mostly impacted by babble noises and is also modulated by the availability of visual speech and the hemisphere (only in babble noise conditions for phrasal CTS and in all noise conditions for syllabic CTS). These observations guided the elaboration of eight relevant features (contrasts) of nCTS in SiN conditions (see S3 Methods): the global level of nCTS and its informational, visual, and hemispheric modulations all for phrasal and syllabic nCTS. In the next sections, we unravel the associations between these features, reading abilities, and classical behavioral predictors of reading. Note the absence of circularity in this approach because features of nCTS were not selected based on their relation with behavioral scores [68]. And on a technical note, seeking association with a limited set of features of nCTS rather than with all nCTS values (32 = 4 noise conditions × 2 visual conditions × 2 hemispheres × 2 frequency ranges of interest) was necessary to avoid introducing close-to-collinear regressors in subsequent analyses and to decrease random errors on nCTS estimates.

### What is the nature of the information about reading abilities brought by measures of SiN processing and classical behavioral predictors of reading?

Having identified relevant features of cortical SiN processing, we first evaluated to which extent these features and classical behavioral predictors of reading bring information about reading abilities in a single, statistically controlled analysis. More precisely, we used partial information decomposition (PID) to dissect the information about reading abilities (target)

brought by behavioral scores (first set of explanatory variables) and features of the nCTS in noise (second set of explanatory variables) [69,70]. Generally speaking, PID can reveal to which extent two sets of explanatory variables bring unique information about a target (information present in one set but not in the other), redundant information (information common to the two sets), and synergistic information (information emerging from the interaction of the two sets). Here, the target consisted of five reading scores: (1) an accuracy and (2) a speed score for the reading of a connected meaningless text (Alouette test) and scores (number of correctly read words per unit of time) for the reading of a list of (3) irregular words, (4) regular words, and (5) pseudowords. The first set of explanatory variables, i.e., the classical behavioral predictors of reading, consisted of a total of five measures indexing phonological awareness (scores on phoneme suppression and fusion tasks), phonological memory (scores on forward and backward digit repetition), and RAN score. The second set of explanatory variables was the eight features of nCTS in SiN conditions identified in the previous subsection. Again, in that analysis, all measures were corrected for age, time spent at school, and IQ (see S2 Methods). For statistical assessment and conversion into easily interpretable $z$-scores, measures of information were compared to the distribution of these measures obtained after permuting reading scores across subjects (see S4 Methods).

As a result, features of nCTS in noise brought significant unique information about reading abilities (unique information, $z = 2.52$; $p = 0.013$), whereas classical behavioral predictors did not (unique information, $z = 1.51$; $p = 0.077$). Both sets of explanatory variables brought significant redundant but not synergistic information about reading (redundant information, $z = 4.22$; $p = 0.0007$; synergistic information, $z = 0.68$; $p = 0.22$).

Further supporting the result that features of nCTS bring significant unique information about reading, this information measure was significantly higher than its permutation distribution in which features of nCTS (rather than reading scores) were permuted across subjects ($p = 0.009$); and so was the value of redundant information ($p = 0.004$). Of notice, the unique information about reading brought by classical behavioral predictors was significantly higher when classical behavioral predictors were not permuted across subjects than when they were ($p = 0.040$); and so was the value of redundant information ($p = 0.010$).

These results indicate that the way the CTS is impacted by ambient noise relates to reading abilities in a way that is not fully explained by classical behavioral predictors of reading. Further analyses will therefore strive to identify which aspects of SiN processing and reading are related and which of these relations are mediated by classical behavioral predictors of reading.

## Which features of SiN processing relate to reading abilities in a way that is not mediated by classical behavioral predictors of reading?

Having identified relevant features of cortical SiN processing, we evaluated to which extent these features bring information about reading abilities above and beyond that provided by classical behavioral predictors of reading. In practice, we identified with linear mixed-effects modeling (1) the set of classical behavioral predictors of reading that best explains reading abilities and (2) the set of features of nCTS in noise that brings additional information about reading abilities. Importantly, all measures were corrected for age, time spent at school, and IQ and were standardized. In that analysis, the type of reading score used to assess reading abilities was taken as a factor. Classical behavioral predictors of reading (five measures) were first entered as regressors before considering the features of nCTS in noise (eight measures) as additional regressors.

Table 3 presents the final linear mixed-effects model fit to reading scores. It shows that RAN score and phonological memory (indexed by the forward digit span) relate to global

**Table 3. Regressors included in the final linear mixed-effects model fit to the five reading scores (dependent variables).** Regressors are listed in their order of inclusion.

| Regressors | $\mathcal{X}^2$ | | $p$ |
|---|---|---|---|
| | df | value | |
| RAN | 1 | 15.8 | <0.0001 |
| Forward digit span | 1 | 11.1 | 0.0009 |
| Visual modulation in phrasal nCTS | 1 | 4.85 | 0.028 |
| Informational modulation in phrasal nCTS dependent on reading score | 5 | 15.6 | 0.0080 |
| Visual modulation in phrasal nCTS dependent on reading score | 4 | 11.1 | 0.026 |

Abbreviations: df, degrees of freedom; nCTS, normalized cortical tracking of speech; RAN, rapid automatized naming

reading abilities. It also shows that two aspects of SiN processing, the visual and informational modulations in phrasal nCTS, explain a different part of the variance in reading abilities. Importantly, these two indices relate to reading in a way that depends on the type of reading score. These effects are illustrated with simple Pearson correlations in Table 4. The time necessary to fulfil the RAN task was significantly negatively correlated with all reading scores. The forward digit span was significantly positively correlated with all reading scores. The visual modulation in phrasal nCTS was overall positively correlated with scores involving reading speed (Alouette speed score and regular, irregular, and pseudoword reading scores; significantly so for pseudoword reading only) but not with the Alouette accuracy score. The informational modulation in phrasal nCTS was characterized by a significant positive correlation with the score on irregular word reading only. Interestingly, the correlation was not significant—and even negative—with the score on pseudoword reading.

We will now attempt to better understand the meaning of this last association (between the informational modulation in phrasal nCTS and irregular but not pseudoword reading). Given that different routes support reading of irregular words (lexical route) and pseudowords (sublexical route), the contrast between corresponding standardized scores (irregular − pseudowords) indicates reading strategy. We henceforth refer to this index as the reading strategy index. Further strengthening the correlation pattern highlighted above for the informational modulation in phrasal nCTS, this latter index correlated even more strongly with the reading strategy index ($r = 0.44$, $p < 0.0001$; see Fig 3, left) than with the score on irregular word

**Table 4. Pearson correlation between measures of reading abilities and relevant brain and behavioral measures.**

| Reading parameter | RAN | Forward digit span | Visual modulation in phrasal nCTS | Informational modulation in phrasal nCTS | Visual modulation in syllabic nCTS | Phoneme suppression | Phoneme fusion |
|---|---|---|---|---|---|---|---|
| Alouette accuracy | −0.37** | 0.33** | 0.00 | −0.03 | 0.29* | 0.11 | 0.25* |
| Alouette speed | −0.41*** | 0.38*** | 0.21# | 0.08 | 0.30** | 0.31** | 0.30** |
| Irregular words | −0.35** | 0.42*** | 0.18 | 0.26* | 0.37** | 0.21# | 0.17 |
| Regular words | −0.42*** | 0.35** | 0.18 | 0.12 | 0.31** | 0.25* | 0.19 |
| Pseudowords | −0.34** | 0.30* | 0.31** | −0.07 | 0.23* | 0.21# | 0.11 |

***$p < 0.001$

**$p < 0.01$

*$p < 0.05$

#$p < 0.1$.

Abbreviations: nCTS, normalized cortical tracking of speech; RAN, rapid automatized naming

Relation between the reading strategy index and phrasal nCTS

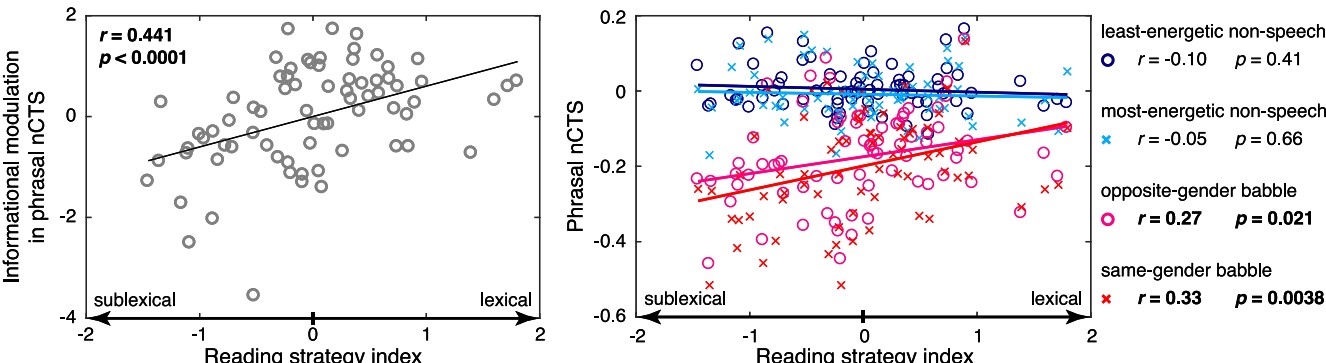

**Fig 3. Relation between the reading strategy index and the nCTS at phrasal rate.** Left—the informational modulation in phrasal nCTS as a function of the reading strategy index. Gray circles depict participants' values, and a black trace is the regression line, with correlation and significance values indicated in the top-left corner. Right—the mean nCTS across visual conditions and both hemispheres for the four types of noise: least-energetic nonspeech (blue circles), most-energetic nonspeech (turquoise crosses), opposite-gender babble (red circles), and same-gender babble (pink crosses). Circles and crosses depict participants' values, and full traces are the regression lines. Correlation and significance level for all noise conditions are indicated on the right of each plot. S3 Data contains the underlying data for this figure. nCTS, normalized cortical tracking of speech.

reading. This suggests that irregular and pseudoword reading scores bring synergistic information about the informational modulation in phrasal nCTS. To confirm this, we used PID to dissect the information about the informational modulation in phrasal nCTS (target) brought by irregular reading scores (first explanatory variable) and pseudoword reading scores (second explanatory variable). This analysis revealed that the score on irregular word reading carried significant, unique information about the informational modulation in phrasal nCTS (unique information, $z = 4.92$, $p = 0.0052$)—whereas the score on pseudowords did not (unique information, $z = -0.21$, $p = 0.38$)—and most interestingly, that these two reading scores carried significant synergistic but not redundant information about the informational modulation in phrasal nCTS (redundant information, $z = -0.55$, $p = 0.63$; synergistic information, $z = 9.73$, $p = 0.0003$).

Fig 3 (right panel) further illustrates that the reading strategy index was correlated with phrasal nCTS only in the babble noise conditions.

In summary, classical behavioral predictors of reading were informative about global reading abilities (similar correlation with all five measures of reading), whereas two aspects of the CTS in noise (informational and visual modulations in phrasal nCTS) related to specific aspects of reading (correlation with some but not all five measures of reading). The extent to which visual speech boosts phrasal CTS in noise was related to reading speed but not accuracy, and the ability to maintain adequate phrasal CTS in babble noise related to reading strategy (dominant reliance on the lexical rather than sublexical route).

### Do other features of SiN processing or classical behavioral predictors of reading relate to reading abilities?

Above, we have identified a set of brain and behavioral measures related to reading. Importantly, each measure was included because it explained a new part of the variance in reading abilities. But the first PID analysis revealed that brain and behavioral measures do carry significant redundant information. This means that some measures might have been left aside if they explained some variance that was already explained (i.e., if they provided mainly redundant information). Accordingly, we also ran the linear mixed-effects analysis with nCTS and behavioral regressors that were not included. This analysis identified an overall positive correlation

between reading abilities and (1) the visual modulation in syllabic nCTS ($\mathcal{X}^2(1) = 9.74$, $p = 0.0018$), (2) phoneme suppression ($\mathcal{X}^2(1) = 4.94$, $p = 0.026$), and (3) phoneme fusion ($\mathcal{X}^2(1) = 4.00$, $p = 0.038$). Corresponding Pearson correlation coefficients are presented in Table 4. A detailed PID analysis revealed that these "side" measures were redundant—and synergistic to some extent—with RAN and forward digit span but not with visual and informational modulations in phrasal nCTS (see S2 Results, S3 Table, and S4 Table). Importantly, these results clarify why behavioral predictors of reading did not bring significant unique information about reading abilities: most of the variance in reading abilities they could explain (maximum $|r| = 0.42$; see Table 4) was also explained by the visual modulation in syllabic nCTS (maximum $|r| = 0.37$). And conversely, the visual modulation in syllabic nCTS was not retained in the final linear mixed-effects model of reading abilities for the same reason.

In summary, scores indexing phonological awareness (score on phoneme suppression and phoneme fusion) and the extent to which visual speech boosts syllabic CTS in noise (visual modulation in syllabic nCTS) relate to global reading abilities in a way that is mediated by the main classical behavioral predictors of reading we identified (RAN and forward digit span) but not with visual and informational modulations in phrasal nCTS.

### Does phonological awareness mediate SiN perception capacities?

Having identified three relations between various aspects of cortical SiN processing and reading, we now specifically test the hypothesis that each of these relations is mediated by phonological awareness. For that, we again relied on PID to decompose the information about reading abilities (target) brought by each identified feature of the CTS in noise (first explanatory variable) and the mean of the two scores indexing phonological awareness (second explanatory variable). Ensuing results are provided in S2 Table. In summary, phonological awareness mediated one aspect of the relation between reading and cortical SiN processing (relation with the benefit of visual speech to boost syllabic CTS in noise) but not the two others (relations involving phrasal CTS in noise).

### Is SiN comprehension accounted for by the features of nCTS related to reading?

If the three features of nCTS related to reading abilities are to index relevant aspects of cortical SiN processing, we would expect them to directly relate to SiN comprehension. To substantiate this consideration, we correlated these features of nCTS with a comprehension score computed as the percentage of correct answers to a total of 40 yes/no forced-choice questions. Again, all variables were corrected for age, time spent at school, and IQ. All three correlations were positive, but none of them were deemed significant (informational modulation in phrasal nCTS, $r = 0.16$, $p = 0.17$; visual modulation in phrasal nCTS, $r = 0.20$, $p = 0.082$; visual modulation in syllabic nCTS, $r = 0.09$, $p = 0.47$). The weakness of these associations could however be explained by ceiling effects in comprehension score due to comprehension questions being too simple. Indeed, 48% of the participants score 38/40 or more.

### Do relations between reading and features of nCTS translate to alterations in dyslexia?

We next evaluated whether the relations between features of nCTS and reading abilities translate to alterations in dyslexia. That analysis was conducted on a group of 26 children with dyslexia and on groups of 26 age-matched and 26 reading-level–matched typically developing children selected among the 73 children included in the first part of the study.

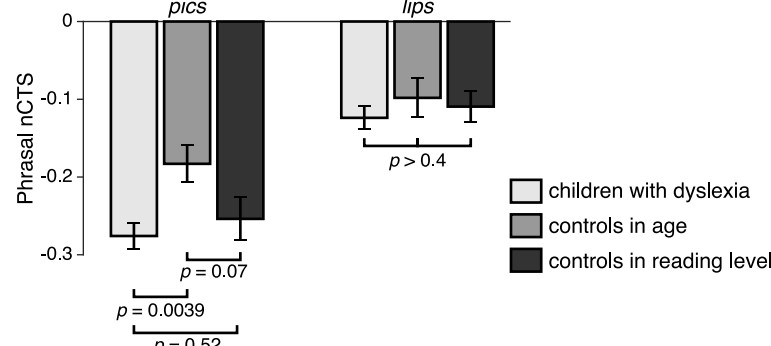

A. Modulations involving phrasal nCTS in dyslexia

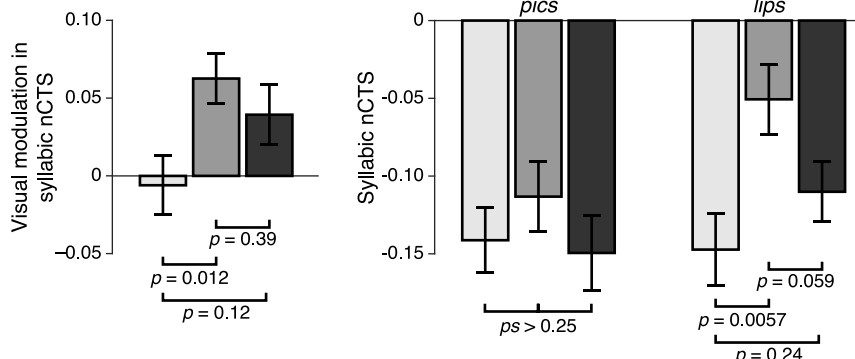

B. Modulations involving syllabic nCTS in dyslexia

**Fig 4. Comparison between children with dyslexia and controls in the measures of nCTS significantly related to reading abilities.** (A) Modulations involving phrasal nCTS. Displayed are the mean and SEM within groups (dyslexia, control in age, and control in reading level) of phrasal nCTS in the conditions with (lips) and without (pics) visual speech information. Values of nCTS were averaged across hemispheres and babble noise conditions for phrasal nCTS and across hemispheres and all noise conditions for syllabic nCTS. (B) Modulations involving syllabic nCTS. On the left is the visual modulation in syllabic nCTS. The right part is as in (A). S4 Data contains the underlying data for this figure. nCTS, normalized cortical tracking of speech.

S5 Table presents the percentage of the 26 children of each reading group (children with dyslexia, controls in age, and controls in reading level) showing statistically significant phrasal and syllabic CTS in each condition. All children showed significant phrasal CTS in all conditions except for one control in age that lacked significant CTS in one of the most challenging conditions (gender-matched babble noise without visual speech information). Qualitatively, fewer controls in reading level (than children with dyslexia and controls in age) showed significant syllabic CTS in all conditions. Still, the percentage of significant CTS remained above 80%, except for controls in reading level in the most-challenging noise conditions (gender-matched babble noises), which indicates that CTS could be robustly assessed at the subject level in all reading groups.

Based on the result that reading abilities relate to phrasal nCTS in babble noise and to the boost in nCTS brought by visual speech, we focused the comparison on the phrasal nCTS in lips and pics averaged across hemispheres and babble noise conditions (see Fig 4A). As a result, phrasal nCTS in pics was similar among individuals with dyslexia and controls in reading level and higher in controls in age (significantly only for children with dyslexia; marginally for controls in reading level). In contrast, phrasal nCTS in lips was similar in all reading groups.

Based on the result that reading abilities relate to the visual modulation in syllabic nCTS, we focused the comparison on this index (see Fig 4B, left part). This revealed that individuals

with dyslexia had significantly lower visual modulation in syllabic nCTS than age-matched but not reading-level–matched controls; the two latter groups showing similar level of visual modulation in syllabic nCTS. To better understand the nature of this difference, we further compared between groups the syllabic nCTS in lips and pics averaged across hemispheres and noise conditions (see Fig 4B, right part). As a result, syllabic nCTS in pics was similar in all reading groups, whereas in lips, it was similar among individuals with dyslexia and controls in reading level and higher in controls in age (significantly for children with dyslexia; marginally for controls in reading level).

In summary, one aspect of cortical SiN processing (reliance on visual speech to boost phrasal nCTS) was not altered in dyslexia, whereas two other aspects (phrasal nCTS in babble noise and reliance on visual speech to boost syllabic nCTS) were altered in dyslexia in comparison with typical readers matched for age but not reading level. This suggests that these two later aspects are altered as a consequence of reduced reading experience.

## Are features of nCTS related to the importance of reading difficulties in dyslexia?

In S3 Results (complemented by S2 Fig), we show that our group with dyslexia was homogenous in terms of reading profile but not in the severity of the reading deficit. This raises the important question of whether and how the reading deficit in dyslexia relates to nCTS in noise. In S4 Results (complemented by S3 Fig, S6 Table, S7 Table, and S8 Table), we answer this question with the same linear mixed-effects modeling approach used in typical readers. However, the results are best illustrated by Pearson correlation between reading scores and nCTS in babble noise conditions in pics and lips (all measures corrected for age, time spent at school, and IQ).

Most surprisingly, phrasal nCTS both in lips and pics for children with dyslexia correlated significantly negatively with all reading scores indexing reading speed but not accuracy or strategy (see Fig 5 and S9 Table). That is, the higher the phrasal nCTS, the slower they read.

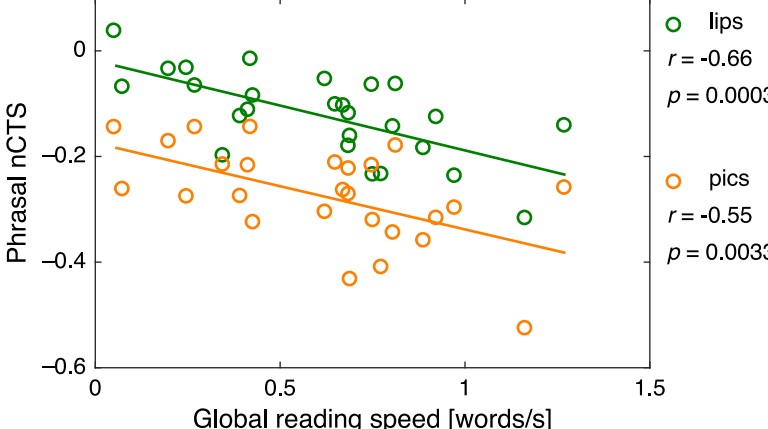

**Fig 5. Relation between reading speed and the nCTS at phrasal rate in dyslexia.** On the x-axis is the mean of the four reading scores indexing reading speed: reading score for irregular, regular, and pseudowords and Alouette reading speed (converted to a number of words read per second). On the y-axis is the mean nCTS across babble noise conditions and both hemispheres for the two types of visual input: pics (orange) and lips (green). Circles depict participants' values, and full traces are the regression lines. Correlation and significance level are indicated on the right. S5 Data contains the underlying data for this figure. nCTS, normalized cortical tracking of speech.

Beyond that, S4 Results show that the informational modulation in phrasal nCTS correlated positively with the difference between reading accuracy and reading speed ($r = 0.51$; $p = 0.0081$). Syllabic nCTS in lips or pics for children with dyslexia did not correlate significantly with any of the reading scores (see S9 Table).

## Discussion

The main objective of this study was to fully characterize the nature of the relation between objective cortical measures of SiN processing and reading abilities in elementary school children. Results demonstrate that some cortical measures of SiN processing relate to reading performance and reading strategy. First, phrasal nCTS in babble (i.e., informational) noise relates to the ability to read irregular but not pseudowords, which in the dual-route cascaded model indicates maturation of the lexical route. Second, the ability to leverage visual speech to boost phrasal nCTS in babble noise relates to reading speed (but not accuracy). Third, the ability to leverage visual speech to boost syllabic nCTS in noise relates to global reading abilities. Fourth, classical behavioral predictors of reading abilities (RAN, phonological memory, and phonological awareness) relate to global reading performance and not strategy. Importantly, behavioral scores and the two features of phrasal CTS in babble noise explained a different part of the variance in reading abilities. Finally, the features of nCTS underlying the first and third relations uncovered in typical readers (phrasal nCTS in babble noise and visual modulation in syllabic nCTS) were significantly altered in dyslexia in comparison with aged-matched but not reading-level–matched typically developing children. However, within the population with dyslexia, nCTS measures of the ability to deal with babble noise were negatively related to reading speed and positively related to the compromise between reading precision and reading speed.

Significant associations were found between reading abilities and some features of phrasal and syllabic nCTS. There is evidence that CTS at phrasal rate (here taken as 0.2–1.5 Hz) partly reflects parsing or chunking of words, phrases, and sentences [71]. Indeed, the brain tracks phrase and sentence boundaries even when speech is devoid of prosody but only if it is comprehensible [41], and the phase of brain oscillations below 4 Hz modulates perception of ambiguous sentences [39]. CTS at phrasal/sentential rate would help align neural excitability with syntactic information to optimize language comprehension [38]. In contrast, CTS at syllable rate (here taken as 2–8 Hz) would reflect low-level auditory processing [71]. In light of the above, our results highlight that associations between SiN perception and reading abilities build on their shared reliance on both language processing and low-level auditory processing.

### Robustness of cortical speech representation to babble noise indexes the degree of development of the lexical route

Our results indicate that an objective cortical measure of the ability to deal with babble noise relates to the maturation of the lexical route. Technically, the informational modulation in phrasal nCTS correlated significantly positively with the reading score on irregular but not pseudowords. Reading score on irregular words indeed provided unique information about the informational modulation in nCTS. Also, the two reading scores in synergy provided some additional information about the informational modulation in nCTS. Furthermore, the result that the informational modulation in nCTS correlated more with the reading strategy index than the score on the irregular words suggests that the key elements at the basis of this relation are the processes needed to read irregular words that are not needed to read pseudowords.

The relation between the degree of development of the lexical route and the level of phrasal nCTS in babble noise could be explained by a positive influence of good SiN abilities on reading acquisition. Let us take as an example the situation of being faced for the first time with a

written word that is read by a teacher while some classmates are making noise. SiN abilities will naturally determine the odds of hearing that word properly and hence the odds of building up the orthographic lexicon. When again reading the word alone, only children with good SiN abilities will have the opportunity to train their lexical route for that specific word. Of course, the same chain of action could be posited for the training of grapheme–phoneme correspondence. But there are many more words than phonemes and syllables, so good SiN abilities might be more important to successfully learning the correspondence between irregular words' orthographic and phonological representations. Indeed, grapheme–phoneme correspondence is intensively trained when learning to read. Children are repeatedly exposed to examples of successful grapheme–phoneme correspondence, some with noise and some without noise. Accordingly, no matter what children's SiN abilities are, they will learn the grapheme–phoneme correspondence and develop their sublexical route provided that they have adequate phonological awareness. Supporting this, phonological awareness does not predict SiN abilities in typical readers [21].

Alternatively, the relation between the ability to read irregular words (which tags the degree of development of the lexical route) and nCTS in babble noise could be mediated by the degree of maturation of the mental lexicon [72,73]. The mental lexicon integrates and binds the orthographic, semantic, and phonological representations of words. Its proper development is important for reading acquisition. Indeed, reading acquisition entails creating a new orthographic lexicon and binding it to the preexisting semantic and phonological lexicons [74]. Development of such binding (1) is indispensable for reading irregular words [75], (2) benefits reading of regular words, and (3) does not contribute to reading pseudowords. The proper degree of development of the mental lexicon is also important for SiN comprehension. Indeed, SiN comprehension strongly depends on lexical knowledge [21,76–78]. And the level of CTS in noise relates to the listeners' level of comprehension [37,42,43]. This therefore suggests that the robustness of CTS to babble noise depends on the level of comprehension, which in turn depends on how developed the mental lexicon is. The degree of development of the mental lexicon could therefore be the hidden factor mediating the relation between SiN and lexical reading ability. This is also perfectly in line with our result that altered phrasal nCTS in babble noise in dyslexia may result from reduced reading experience. In brief, reading difficulties in dyslexia would reduce their reading experience, which would impair building up the mental lexicon and in turn impede SiN perception. Still, future studies on the association between SiN processing and reading should include measures of the degree of development of the mental lexicon to carefully analyze the interrelation between SiN perception, reading abilities, and the degree of development of the mental lexicon.

Our results in dyslexia support the existence of a relation between reading abilities and cortical measures of the ability to deal with SiN, but they bring important nuances. First, phrasal nCTS in nonvisual babble noise conditions was altered in children with dyslexia compared with age-matched but not reading-level–matched controls, indicating that such alteration could be due to variability in reading experience. Second, within the children with dyslexia, phrasal nCTS was globally and negatively correlated with reading speed, and the informational modulation in phrasal nCTS was positively correlated with the contrast between reading accuracy and reading speed. These two relations could be explained by compensatory attentional mechanisms so that children with severe dyslexia developed enhanced attentional abilities at the basis of improved SiN abilities and more accurate—despite still slower—reading (compared with children with a mild dyslexia). Hence, such relations might hold only in children with dyslexia free of attentional disorder, as was the case with our participants. Also, it should be remembered that these relations were found in a relatively small sample of children with dyslexia ($n = 26$) and should be confirmed by future studies.

## Audiovisual integration and reading abilities

We found significant relations between reading abilities and the ability to leverage visual speech to maintain phrasal and syllabic CTS in noise. Visual speech cues (articulatory mouth and facial gestures) are well known to benefit SiN comprehension [61] and CTS in noise [79–83]. Obviously, the auditory signal carries much more fine-grained information about the phonemic content of speech than the visual signal. But the effect of audiovisual speech integration is quite evident in SiN conditions, in which it affords a substantial comprehension benefit [61,62,84,85]. Mirroring this perceptual benefit, it is already well documented that phrasal and syllabic CTS in noise is boosted in adults when visual speech information is available [79–83,86–89].

We found that the visual modulation in phrasal nCTS correlated globally and positively with reading speed (significantly so for the pseudowords) but not accuracy. However, our children with dyslexia (compared with both control groups) did not have any alteration in their phrasal nCTS in babble noise when visual speech was provided. Instead, they successfully relied on visual speech information to restore their phrasal CTS in babble noise (which was altered without visual speech information). In other words, reliance on lipreading to maintain appropriate phrasal CTS in babble noise appeared as a protection factor in our group of children with dyslexia.

We also found that the visual modulation in syllabic nCTS correlated globally and positively with reading abilities. More interestingly, our children with dyslexia (compared with both control groups) did not have any significant alteration in their syllabic nCTS in noise when visual speech was not provided. However, compared with age-matched typically developing children, they benefited significantly less from visual speech to boost syllabic CTS in noise. Instead, they behaved more like reading-level–matched typically developing children. Accordingly, our results cannot argue against the view that poor audiovisual integration in dyslexia is caused by reduced reading experience [63,90,91]. Notwithstanding, the pattern of results (see Fig 4B left) is even suggestive of an alteration in dyslexia in comparison with reading-level–matched children. More statistical power would be needed to confirm/disprove the trend.

Our result that audiovisual integration abilities correlate with reading abilities is in line with existing literature. Indeed, individuals with dyslexia benefit less from visual cues to perceive SiN than typical readers [92–96]. Audiovisual integration and reading could be altered in dyslexia simply because both rely on similar mechanisms. Indeed, reading relies on the ability to bind visual (graphemic) and auditory (phonemic) speech representations [97,98]. And according to some authors, suboptimal audiovisual integration mechanisms could reduce reading fluency [99]. Importantly, the finding that individuals with dyslexia benefit normally from visual speech to boost phrasal but not syllabic CTS in noise brings important information about the nature of the audiovisual integration deficit in dyslexia. Following the functional roles attributed to CTS, individuals with dyslexia would properly integrate visual speech information to optimize processing of syntactic information [38] but not to support acoustic/phonemic processing [71]. This could be explained by their preserved ability to extract and integrate the temporal dynamics of visual speech but not the lip configuration [96], two aspects of audiovisual speech integration currently thought to be supported by distinct neuronal pathways [100]. This inability to rely on lip configuration to improve auditory phonemic perception in SiN conditions may be caused by a supramodal phonemic categorization deficit, as already proposed for children with specific language impairment [101]. Finally, the fact that the visual modulation in syllabic nCTS brought a limited amount of unique information about reading with respect to classical behavioral predictors of reading, but that all of them brought

more information in synergy, suggests that a broad set of low-level processing abilities contribute to determining reading abilities and alterations in dyslexia [102,103].

## Classical behavioral predictors related to global reading abilities

Our results confirm that classical behavioral predictors of reading (RAN, phonological memory, and metaphonological abilities) are directly related to the global reading level rather than reading strategy. We draw this conclusion because the optimal model for reading score contained a common slope for all reading subtests. This means that the model was not significantly improved by optimizing the slope for each of the five reading subtests separately. Accordingly, univariate correlation coefficients presented in Table 4 were roughly similar across the five reading scores.

Phonological memory (assessed with forward digit span) was significantly positively correlated with the global reading level. That phonological memory relates to global reading abilities rather than reading strategy is well documented [4]. Poor readers, regardless of their reading profile, typically perform poorly on phonological memory tests involving digits, letters [104,105], or words [106].

Performance on the RAN task was also related to the global reading level, in line with existing literature [6–8,107–110]. RAN performance indeed has a moderate to strong relationship with all classical reading measures alike, including word, nonword, and text reading, as well as text comprehension [107]. It is a consistent predictor of reading fluency in various alphabetic orthographies independent of their complexity [111]. RAN performance even predicts reading performance similarly well at an interval of 2 years [112] for reading performance assessed with tasks tagging lexical and sublexical routes. It is thought that RAN and reading performances correlate because they involve serial processing and oral production [110], two processes that are common to both reading routes.

Finally, phonological awareness assessed with phoneme suppression and fusion tasks was significantly related to reading abilities. However, the information it brought about reading was less and essentially redundant with that brought by RAN and phonological memory. This is not surprising given that children tested in the present study had at least 1 year of reading experience. Phonological awareness indeed plays a key role in the early stages of reading acquisition, i.e., when learning grapheme-to-phoneme conversion [113–115], and undergoes a substantial maturation during that period [116].

## Phonological awareness

Our results indicate that, in typical readers, phonological awareness mediates at best part of the relation between the cortical processing of SiN and reading abilities. Indeed, the information about reading brought by phonological awareness was redundant with that brought by the visual modulation in syllabic nCTS but not with that brought by the informational and visual modulations in phrasal nCTS. This finding illustrates the importance of separating the different processes involved in SiN processing and reading to seek associations. It also provides a potential reason why contradictory reports exist on the topic [19–21].

Nevertheless, the role of phonological awareness might have been underestimated in the present study because of a lack of sensitivity in our phonological awareness subtests. Indeed, phonological awareness tasks turned out to be too easy for older participants, leading to ceiling effects (about half of the participants reached the maximum score on phoneme fusion and suppression tasks). This could explain the weak relation observed between reading abilities and phonological awareness skills. In contrast, there was no ceiling effect for the RAN, which may explain the strong correlation between this score and reading abilities.

### Further discussion

In S1 Discussion, we discuss considerations related to the fact that (1) only one acoustic signal-to-noise ratio was studied, (2) regression models to estimate CTS in a given condition were trained on all other conditions, (3) occipital sensors were included in regression models to estimate CTS, and (4) the study was conducted in French. We also discuss the potential yield of future studies in illiterate adults.

### Conclusion

Overall, these results significantly further our understanding of the nature of the relation between SiN processing abilities and reading abilities. They demonstrate that cortical processing of SiN and reading abilities are related in several specific ways and that some of these relations translate into alterations in dyslexia that are attributable to reading experience. However, within children with dyslexia, these relations appeared changed or even reversed, potentially owing to compensatory attentional mechanisms. Our results also demonstrate that classical behavioral predictors of reading (including phonological awareness) mediate relations involving the processing of acoustic/phonemic but not syntactic information in natural SiN conditions. This contrasts with the classically assumed mediating role of phonological awareness. Instead, the ability to process speech syntactic content in babble noise (indexed by phrasal nCTS) could directly modulate skilled reading acquisition. Finally, the information about reading abilities brought by cortical markers of syntactic processing of SiN was complementary to that provided by classical behavioral predictors of reading. This implies that such markers of SiN processing could serve as novel electrophysiological markers of reading abilities.

### Methods

#### Participants

In total, 73 typical readers (mean ± SD age, 8.74 ± 1.41 years; age range, 6.70–11.72 years) and 26 children with dyslexia (mean ± SD age, 10.24 ± 1.08 years; age range, 7.97–12.29 years) enrolled in elementary school took part in this experiment (see Table 1 for participants' characteristics). Children with dyslexia had received a diagnosis of dyslexia, which implies that children had (at the time of diagnosis) at least 2 years of delay in reading acquisition that could not be explained by low IQ or social or sensitive disorders. All were native French speakers, reported being right-handed, had normal hearing according to pure-tone audiometry (normal hearing thresholds between 0–25 dB HL for 250, 500, 1,000, 2,000, 4,000, and 8,000 Hz) and normal SiN perception as revealed by a SiN test (Lafon 30) from a French language central auditory battery [117]. We used a French translation of the Family Affluence Scale [118] to evaluate participants' socioeconomic level.

This study was approved by the local ethics committee (Comité d'Ethique Hospitalo-Facultaire Erasme-ULB, 021/406, Brussels, Belgium; approval number: P2017/081) and conducted according to the principles expressed in the Declaration of Helsinki. Participants were recruited mainly from local schools through flyer advertisements or from social networks. Participants and their legal representatives signed a written informed consent before participation. Participants were compensated with a gift card worth 50 euros.

#### Behavioral assessment

Participants underwent a comprehensive behavioral assessment intended to appraise their reading abilities and some cognitive abilities related to reading or speech perception.

**Reading abilities.** Children completed the word-reading (regular, irregular, and pseudo-words) tasks of a dyslexia detection tool (ODEDYS-2; [119] and the Alouette-R reading task [120]).

For each of the word-reading tasks (regular, irregular, or pseudowords), participants had to read as rapidly and accurately as possible a list of 20 words. Each task provided a reading score computed as the number of words correctly read divided by the reading time (in seconds).

In the Alouette-R task [120], children had 3 min to read as rapidly and accurately as possible a text of 256 words. This text is composed of a succession of words that do not tell a meaningful story. This peculiarity forces children to solely rely on their reading skills and prevents children from using anticipation or inference strategies that could boost the reading scores. An accuracy score was computed as the number of words correctly read divided by the total number of words read, and a speed score was computed as the number of words correctly read multiplied by the ratio of 180 s (maximal reading time) to the effective reading time.

**Phonological processing.** The initial phoneme suppression and initial phonemes fusion tasks of the ODEDYS-2 [119] were used to assess phonological processing.

In the initial phoneme suppression task, children had to repeat orally presented words while intentionally suppressing the initial phoneme of the word (i.e., dog → og). In total, 10 words were presented, and performance was quantified as the percentage correct.

In the initial phoneme fusion task, children had to combine the initial phoneme of two orally presented words to create a new (non-)word (i.e., Big & Owen → /bo/). In total, 10 pairs of words were presented, and performance was quantified as percentage correct.

**RAN.** We used the RAN task of the ODEDYS-2 [119]. Children had to name as rapidly and accurately as possible 25 pictures (five different pictures randomly repeated five times). Performance was quantified as the total time to complete the task, meaning that the lower the score, the better the performance.

**Phonological memory.** The forward and backward digit repetition task from the ODE-DYS-2 [119] was used to assess phonological memory.

In the forward digit repetition task, children were asked to repeat orally presented number series in the same order as presented. The series are different at every trial. The first series contains three digits, and the size of the series is incremented by one every second trial. The task ends after a failure to repeat the two series of a given size. Forward digit span score was taken as the number of digits in the last correctly repeated series.

The backward digit repetition task is akin to the forward one. The only difference is that digit series have to be repeated in the exact reverse order (e.g., children presented 1 2 3 4 have to repeat 4 3 2 1).

**Attention abilities.** The bells test [121] was used to assess visual attention, and the TAP auditory attention subtest [122] was used to assess the auditory attentional level.

In the bells test, children had 2 min to find as many bells as possible on a sheet comprising 35 bells scattered among 280 visual distractors. Performance was quantified as the number of bells found divided by the time needed.

In the TAP auditory attention subtest, children had to focus their attention during 3 min 20 s on an auditory stream. Children heard a train of 200 pure-tone stimuli lasting 500 ms with a 1,000-ms stimulus-onset asynchrony. Tones alternated between high (1,073 Hz) and low (450 Hz) pitch. There were 16 occurrences in which two high- or low-pitch tones were following one another. Only in this case, participants had to press a response button as fast as possible. A performance score was quantified as the number of correct responses, a speed score as the mean response time, and a failure score as the number of responses to tones differing in pitch with the preceding one.

**Nonverbal intelligence.** The brief version of the Weschler Nonverbal (WNV) Scale of Ability [123] was used to assess nonverbal intelligence.

This assessment consisted of matrices and recognition subtests for children younger than 8 years. Older children were assessed with matrices and spatial memory subtests.

In the matrices subtest, children were presented with incomplete visual matrices and had to select the correct missing portion among four or five response options. The subtest ended when four mistakes were made in the last five trials. A raw score was taken as the number of correctly completed matrices. This raw score was converted to a *T* score by comparison with values provided in a table of norms.

In the recognition subtest, children had to carefully look at visual geometric designs that were presented one by one for 3 s. After each presentation, they had to identify the previously seen design among four or five response options. The subtest ended when four mistakes were made in the last five recognition trials. A raw score was taken as the number of correctly recognized drawings. This raw score was converted to a *T* score by comparison with values provided in a table of norms.

In the spatial memory subtest, children were presented with a board with 10 cubes spread on it and were asked to mimic the examiner's tapping sequence. The sequences are different on every trial. The first sequence consists of tapping on two cubes, and the size of the sequences is incremented by one every second trial. The task ends after a failure to repeat two sequences of a given size. This task was performed twice, in forward and backward directions. For each direction, a raw score was taken as the number of correctly repeated sequences. Raw scores were summed and converted to a *T* score by comparison with values provided in a table of norms.

Total nonverbal IQ was computed as the sum of both *T* scores, which was compared with a table of norms, providing a total nonverbal IQ score.

## Neuroimaging assessment

**Stimuli.** The stimuli were derived from 12 audiovisual recordings of four native French-speaking narrators (two females, three recordings per narrator) telling a story for approximately 6 min (mean ± SD, 6.0 ± 0.8 min) (for more details, see S5 Methods). Fig 1 illustrates the time course of a video stimulus. In each video, the first 5 s were kept unaltered to enable children to unambiguously identify the narrator's voice and face that they were requested to attend to. The remainder of the video was divided into 10 consecutive blocks of equal size that were assigned to nine conditions. Two blocks were assigned to the noiseless condition, in which the audio track was kept but the video was replaced by static pictures illustrating the story (mean ± SD picture presentation time across all videos, 27.7 ± 10.8 s). The remaining eight blocks were assigned to eight conditions in which the original sound was mixed with a background noise at 3 dB signal-to-noise ratio. There were four different types of noise, and each type of noise was presented once with the original video, thereby giving access to lip-read information (lips visual conditions), and once with the static pictures illustrating the story (pics visual conditions). The different types of noise differed in the degree of energetic and informational interference they introduced [57]. Fig 1 and S1 Fig illustrate their spectral and spectrotemporal properties. The least-energetic nonspeech (i.e., noninformational) noise was a white noise high-pass filtered at 10,000 Hz. The most-energetic nonspeech noise had its spectral properties dynamically adapted to mirror those of the narrator's voice approximately 1 s around. It was derived from the actual narrators' audio recording by (1) Fourier transforming the sound in 2-s-long windows sliding by step of 0.5 s, (2) replacing the phase by random numbers, (3) inverse Fourier transforming the Fourier coefficients in each window, (4) multiplying

these phase-shuffled sound segments by a sine window (i.e., half a sine cycle with 0 at edges, and 1 in the middle), and (5) summing the contribution of each overlapping window. The opposite-gender babble (i.e., informational) noise was a five-talker cocktail party noise recorded by individuals of gender opposite to the narrator's (i.e., five men for female narrators). The same-gender babble noise was a five-talker cocktail party noise recorded by individuals of gender identical to the narrator's. For both babble noises, the five individual noise components were obtained from a French audiobook database ([http://www.litteratureaudio.com](http://www.litteratureaudio.com)), normalized, and mixed linearly. The assignment of conditions to blocks was random, with the constraint that each of the five first and last blocks contained exactly one noiseless audio and each type of noise, two with lips videos and two with pics videos. Smooth audio and video transitions between blocks was ensured with 2-s fade-in and fade-out. Ensuing videos were grouped in three disjoint sets featuring one video of each of the narrators (total set duration: 23.0, 24.3, 24.65 min), and there were four versions of each set differing in condition random ordering.

**Experimental paradigm.** During the imaging session, participants lay on a bed with their head inside the MEG helmet. Their brain activity was recorded while they were attending four videos (separate recording for each video) of a randomly selected set and ordering of the videos presented in a random order, and finally while they were at rest (eyes opened, fixation cross) for 5 min. They were instructed to watch the videos attentively, listen to the narrators' voice while ignoring the interfering noise, and remain as still as possible. After each video, they were asked 10 yes/no simple comprehension questions. Videos were projected onto a back-projection screen placed vertically, approximately 120 cm away from the MEG helmet. The inner dimensions of the black frame were 35.2 cm (horizontal) and 28.8 cm (vertical), and the narrator's face spanned approximately 15 cm (horizontal) and approximately 20 cm (vertical). Participants could see the screen through a mirror placed above their head. In total, the optical path from the screen to participants' eyes was of approximately 150 cm. Sounds were delivered at 60 dB (measured at ear level) through a MEG-compatible, front-facing, flat-panel loudspeaker (Panphonics Oy, Espoo, Finland) placed approximately 1 m behind the screen.

**Data acquisition.** During the experimental conditions, participants' brain activity was recorded with MEG at the CUB Hôpital Erasme. Neuromagnetic signals were recorded with a whole-scalp–covering MEG system (Triux, MEGIN) placed in a lightweight, magnetically shielded room (Maxshield, MEGIN), the characteristics of which are described elsewhere [124]. The sensor array of the MEG system comprised 306 sensors arranged in 102 triplets of one magnetometer and two orthogonal planar gradiometers. Magnetometers measure the radial component of the magnetic field, whereas planar gradiometers measure its spatial derivative in the tangential directions. MEG signals were band-pass filtered at 0.1–330 Hz and sampled at 1,000 Hz.

We used four head-position indicator coils to monitor the subjects' head position during the experimentation. Before the MEG session, we digitized the location of these coils and at least 300 head-surface points (on scalp, nose, and face) with respect to anatomical fiducials with an electromagnetic tracker (Fastrack, Polhemus).

Finally, subjects' high-resolution 3D T1-weighted cerebral images were acquired with a magnetic resonance imaging (MRI) scanner (MRI 1.5T, Intera, Philips) after the MEG session.

**Data preprocessing.** Continuous MEG data were first preprocessed off-line using the temporal signal space separation method implemented in MaxFilter software (MaxFilter, MEGIN; correlation limit 0.9, segment length 20 s) to suppress external interferences and to correct for head movements [125,126]. To further suppress physiological artifacts, 30 independent components were evaluated from the data band-pass filtered at 0.1–25 Hz and reduced to a rank of 30 with principal component analysis. Independent components corresponding to

heartbeat, eye-blink, and eye-movement artifacts were identified, and corresponding MEG signals reconstructed by means of the mixing matrix were subtracted from the full-rank data. Across subjects and conditions, the number of subtracted components was 3.45 ± 1.23 (mean ± SD across subjects and recordings). Finally, a window time of 1-s time points at timings 1 s around remaining artifacts were set to bad. Data were considered contaminated by artifacts when MEG amplitude exceeded 5 pT in at least one magnetometer or 1 pT/cm in at least one gradiometer.

We extracted the temporal envelope of the attended speech (narrators' voice) using the optimal approach proposed by Biesmans and colleagues [127]. Briefly, audio signals were band-pass filtered using a gammatone filter bank (15 filters centered on logarithmically spaced frequencies from 150 Hz to 4,000 Hz), and sub-band envelopes were computed using Hilbert transform, elevated to the power 0.6, and averaged across bands.

**Accuracy of speech envelope reconstruction and normalized CTS.** For each condition and participant, a global value of cortical tracking of the attended speech was evaluated for all left-hemisphere sensors at once and for all right-hemisphere sensors at once. Using the mTRF toolbox [64], we trained a decoder on MEG data to reconstruct speech temporal envelope and estimated its Pearson correlation with real speech temporal envelope. This correlation is often referred to as the reconstruction accuracy, and it provides a global measure of CTS. See S6 Methods for a full description of the procedure and statistical assessment. A similar approach has been used in previous studies on the CTS [50,54,66,67].

Based on CTS values, we derived the normalized CTS (nCTS) in SiN conditions as the following contrast between CTS in SiN ($CTS_{SiN}$) and noiseless ($CTS_{noiseless}$) conditions:

$$nCTS = (CTS_{SiN} - CTS_{noiseless})/(CTS_{SiN} + CTS_{noiseless}).$$

Such contrast presents the advantage of being specific to SiN processing abilities by factoring out the global level of CTS in the noiseless condition. However, it can be misleading when derived from negative CTS values (which may happen because CTS is an unsquared correlation value). For this reason, CTS values below a threshold of 10% of the mean CTS across all subjects, conditions, and hemispheres were set to that threshold prior to nCTS computation. Thanks to this thresholding, the nCTS index takes values between −1 and 1, with negative values indicating that the noise reduces CTS.

## PID

All behavioral and nCTS measures were corrected for IQ, age, time spent at elementary school, and outliers (see S2 Methods).

We used PID to appraise without a priori the relation between reading abilities, cortical measures of SiN processing, and classical behavioral predictors of reading. In general, PID decomposes the mutual information (MI) quantifying the relationship between two explanatory variables (or sets of explanatory variables) and a single target into four constituent terms: the unique information about the target, which is available separately from each explanatory variable alone; the redundant or shared information, which is common to the two explanatory variables; and synergistic information, which is information about the target that is available only when both explanatory variables are observed together (e.g., the relationship between their values is informative about the target) [69,70,128]. PID was previously used to decompose the information brought by acoustic and visual speech signals about brain oscillatory activity [80] and to compare auditory encoding models of MEG during speech processing [128]. In our analysis, the five reading scores were used as the target, the features of nCTS as the first set of explanatory variables, and behavioral scores as the second set of explanatory

variables. PID was also used to better understand the nature of some other statistical associations we uncovered. For further details on PID, its quantification with *z*-scores, and its statistical assessment, see S4 Methods.

## Linear mixed-effects modeling of nCTS and reading values

We performed linear mixed-effects analysis with R [129] and *lme4* [130] to identify how different fixed effects modulate nCTS. We started with a null model that included only a different random intercept for each subject. The model was iteratively compared with models incremented with simple fixed effects of hemisphere, noise (least-energetic nonspeech, most-energetic nonspeech, opposite-gender babble, and same-gender babble), and visual (lips versus pics) added one by one. At every step, the most significant fixed effect was retained until the addition of the remaining effects did not improve the model any further ($p > 0.05$). The same procedure was then repeated to refine the ensuing model with the interactions of the simple fixed effects of order 2 (e.g., hemisphere × noise) and then 3 (hemisphere × noise × visual).

We followed the same approach to identify how reading abilities (five standardized scores) relate to classical behavioral predictors of reading and features of nCTS. In that analysis, we first considered a nonzero slope for the classical behavioral predictors identical for all reading scores, then a nonzero slope for the classical behavioral predictors different for all reading scores, then a nonzero slope for the features of nCTS identical for all reading scores, and finally a nonzero slope for the features of nCTS different for all reading scores.

Of note, we preferred linear mixed-effects modeling over other statistical methods for two reasons. (1) This method could identify both the factors that modulate nCTS and the regressors that explain reading scores. (2) It could simultaneously model all the reading scores and identify possible differences in correlation with the different readings scores.

Also worth noting, performing model selection with a stepwise deletion approach (i.e., when starting with the full model and iteratively removing fixed effects that did not decrease significantly model accuracy) yielded the exact same linear mixed-effects models.

## Supporting information

**S1 Methods. Assessment of the degree of energetic masking.**
(DOCX)

**S2 Methods. Preprocessing of brain and behavioral indices.**
(DOCX)

**S3 Methods. Extraction of the relevant features of nCTS.** nCTS, normalized cortical tracking of speech.
(DOCX)

**S4 Methods. Partial information decomposition.**
(DOCX)

**S5 Methods. Recording of video stimuli.**
(DOCX)

**S6 Methods. Accuracy of speech envelope reconstruction.**
(DOCX)

**S1 Results. Contribution of visual cortical activity to nCTS.** nCTS, normalized cortical tracking of speech.
(DOCX)

**S2 Results. Side measures are redundant with RAN and digit span but not with modulations in phrasal nCTS.** nCTS, normalized cortical tracking of speech; RAN, rapid automatized naming.
(DOCX)

**S3 Results. Reading profile and reading deficit in the group with dyslexia.**
(DOCX)

**S4 Results. Are features of nCTS related to the importance of reading difficulties in dyslexia?** nCTS, normalized cortical tracking of speech.
(DOCX)

**S1 Discussion. Supplementary Discussion.**
(DOCX)

**S1 Table. Percentage of the 73 typical readers showing significant CTS at phrasal and syllabic rates in the nine different conditions.** The two values provided for the noiseless condition correspond to two arbitrary subdivisions of the noiseless data to match the amount of data for the eight noise conditions. CTS, cortical tracking of speech.
(DOCX)

**S2 Table. Nature of the information about reading abilities brought by each of the three uncovered features of the CTS in noise and phonological awareness (mean of the scores for phoneme fusion and suppression).** Significant values ($p < 0.05$) are displayed in boldface, and marginally significant values are displayed in boldface and italicized. CTS, cortical tracking of speech.
(DOCX)

**S3 Table. Nature of the information about reading brought by (1) the visual modulation in syllabic nCTS and (2) each of the four regressors included in the final model of reading abilities (informational modulation in phrasal nCTS, visual modulation in phrasal nCTS, forward digit span, and RAN).** nCTS, normalized cortical tracking of speech; RAN, rapid automatized naming.
(DOCX)

**S4 Table. Same as in S3 Table for metaphonological abilities.**
(DOCX)

**S5 Table. Percentage of the 26 children of each reading group (dyslexia, control in age, and control in reading level) showing significant CTS in at least one hemisphere at phrasal and syllabic rates in the nine different conditions.** CTS, cortical tracking of speech.
(DOCX)

**S6 Table. Factors included in the final linear mixed-effects model fit to the nCTS (independent variable) at phrasal and at syllabic rates in children with dyslexia.** Factors are listed in their order of inclusion. nCTS, normalized cortical tracking of speech.
(DOCX)

**S7 Table. Regressors included in the final linear mixed-effects model fit to the five reading scores (dependent variables) in children with dyslexia.** Regressors are listed in their order of inclusion.
(DOCX)

**S8 Table. Pearson correlation between measures of reading abilities and relevant brain and behavioral measures in children with dyslexia.** ***$p < 0.001$, **$p < 0.01$, *$p < 0.05$, #$p < 0.1$. nCTS, normalized cortical tracking of speech.
(DOCX)

**S9 Table. Pearson correlation between measures of reading abilities and nCTS measures in children with dyslexia.** ***$p < 0.001$, **$p < 0.01$, *$p < 0.05$, #$p < 0.1$. nCTS, normalized cortical tracking of speech.
(DOCX)

**S1 Fig.** Spectrogram of a 4-s excerpt of attended speech (A) and corresponding noise (B) in the range of 0–7 kHz. Wide-band spectrograms (0–20 kHz) are also presented for the attended speech and the least-energetic nonspeech noise (C) to show that noise power was confined to frequencies above 10 kHz in this latter noise condition. The zeros of the dBFS were fixed based on the attended speech spectrogram and applied to all noise spectrograms. dBFS, decibel full scale.
(TIF)

**S2 Fig. Relation between reading abilities and the nCTS at phrasal rate in dyslexia.** S6 Data contains the underlying data for this figure. nCTS, normalized cortical tracking of speech.
(TIF)

**S3 Fig.** Impact of the main fixed effects on the nCTS at phrasal (A) and syllabic rates (B) in children with dyslexia. All is as in Fig 2. S7 Data contains the underlying data for this figure. nCTS, normalized cortical tracking of speech.
(TIF)

**S1 video. Exemplary video stimulus wherein static pictures were replaced by text descriptions.**
(M4V)

**S1 data. Behavioral and CTS values for all participants.** CTS, cortical tracking of speech.
(XLSX)

**S2 data. Raw data underlying Fig 2.**
(XLSX)

**S3 data. Raw data underlying Fig 3.**
(XLSX)

**S4 data. Raw data underlying Fig 4.**
(XLSX)

**S5 data. Raw data underlying Fig 5.**
(XLSX)

**S6 data. Raw data underlying S2 Fig.**
(XLSX)

**S7 data. Raw data underlying S3 Fig.**
(XLSX)

## Acknowledgments

We thank Wafae El Hammouchi, Morgane De Boeck, Konstantina Kanellou, and Pauline Delvingt for help with data acquisition.

## Author Contributions

**Conceptualization:** Florian Destoky, Marc Vander Ghinst, Xavier De Tiège, Mathieu Bourguignon.

**Formal analysis:** Florian Destoky, Mathieu Bourguignon.

**Funding acquisition:** Jacqueline Leybaert, Xavier De Tiège, Mathieu Bourguignon.

**Investigation:** Florian Destoky, Julie Bertels, Maxime Niesen, Mathieu Bourguignon.

**Methodology:** Julie Bertels, Vincent Wens, Marie Lallier, Robin A. A. Ince, Joachim Gross, Xavier De Tiège, Mathieu Bourguignon.

**Project administration:** Mathieu Bourguignon.

**Resources:** Xavier De Tiège.

**Supervision:** Xavier De Tiège, Mathieu Bourguignon.

**Writing – original draft:** Florian Destoky, Mathieu Bourguignon.

**Writing – review & editing:** Florian Destoky, Julie Bertels, Maxime Niesen, Vincent Wens, Marc Vander Ghinst, Jacqueline Leybaert, Marie Lallier, Robin A. A. Ince, Joachim Gross, Xavier De Tiège, Mathieu Bourguignon.

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
