## [Editor Report · Decision Letter 0]

9 Feb 2020

Dear Dr Destoky, 

Thank you for submitting your manuscript entitled "Cortical tracking of speech in noise accounts for reading strategies in children" for consideration as a Research Article by PLOS Biology.

Your manuscript has now been evaluated by the PLOS Biology editorial staff, as well as by an Academic Editor with relevant expertise, and I am writing to let you know that we would like to send your submission out for external peer review. Please accept my apologies for the delay in sending this decision to you.

Before we can send your manuscript to reviewers, we need you to complete your submission by providing the metadata that is required for full assessment. To this end, please login to Editorial Manager where you will find the paper in the 'Submissions Needing Revisions' folder on your homepage. Please click 'Revise Submission' from the Action Links and complete all additional questions in the submission questionnaire.

Please re-submit your manuscript within two working days, i.e. by Feb 12 2020 11:59PM.

Kind regards,

Gabriel Gasque, Ph.D.,

Senior Editor

PLOS Biology

---

## [Decision Letter · Decision Letter 1]

10 Apr 2020

Dear Dr Destoky,

Thank you very much for submitting your manuscript "Cortical tracking of speech in noise accounts for reading strategies in children" for consideration as a Research Article at PLOS Biology. Your manuscript has been evaluated by the PLOS Biology editors, by an Academic Editor with relevant expertise, and by three independent reviewers.

The reviews of your manuscript are appended below. You will see that the reviewers find the work potentially interesting. However, based on their specific comments and following discussion with the Academic Editor, I regret that we cannot accept the current version of the manuscript for publication. We remain interested in your study and we would be willing to consider resubmission of a comprehensively revised version that thoroughly addresses all the reviewers' comments. We cannot make any decision about publication until we have seen the revised manuscript and your response to the reviewers' comments. Your revised manuscript would be sent for further evaluation by the reviewers. Please note that a positive outcome is not guaranteed at this stage. 

If you decide to revise for PLOS Biology, your revisions should address the specific points made by each reviewer. As you will see, reviewer 1 raises the fundamental issue of whether SiN tracking does actually reflect performance, reviewer 2 raises the issue of large number of readings and MEG measures and the reproducibility of the correlation and suggests a very reasonable analysis to test on current data set, and reviewer 3 raises issues with respect to sample characterisation. The Academic Editor thinks, and we agree, that all these concerns should be thoroughly addressed for a successful revision.

We appreciate that our and the reviewers’ requests represent a great deal of extra work, and we are willing to relax our standard revision time to allow you six months to revise your manuscript. We expect to receive your revised manuscript within 6 months.

Please email us (plosbiology@plos.org) if you have any questions or concerns, or would like to request an extension --we understand these are particularly complex times. At this stage, your manuscript remains formally under active consideration at our journal; please notify us by email if you do not intend to submit a revision so that we may end consideration of the manuscript at PLOS Biology.

**IMPORTANT - SUBMITTING YOUR REVISION**

As stated above, your revisions should address the specific points made by each reviewer. Please submit the following files along with your revised manuscript:

*Resubmission Checklist*

*Published Peer Review*

*PLOS Data Policy*

*Blot and Gel Data Policy*

Sincerely,

Gabriel Gasque, Ph.D., 

Senior Editor

PLOS Biology

REVIEWS:

Reviewer #1: Review: PBIOLOGY-D-20-00208R1

The manuscript presents an intriguing investigation of the relationship between indicators of reading skill and cortical tracking of speech in noise in the presence of additional visual (lip-movement, 'lips') information or in its absence ('pics'). They examine tracking at two time-scales, the phrasal and the syllabic rate. They compare the modulation of tracking by visual information in the presence of informational and non-informational noise.

As a result of the number of factors investigated and the number of outcomes presented, I found the paper overly dense and the conclusions hard to evaluate. Cortical tracking (nCTS) is evaluated for each hemisphere, for 2 visual conditions, for 4 noise conditions at 2 time-scales. These 32 values are then somehow reduced to 8 "contrasts" which are then related to a an array of reading indicators. This is achieved through a combination of variable selection using LME followed by PID. 

The data point to a relationship between the enhancement of cortical tracking in noise by the provision of supporting visual information and several indicators of reading performance. The authors suggest that the ability to use visual information to enhance cortical entrainment to speech is related to reading ability. One potential mechanism offered is that in noisy classroom situations, children with better ability to track speech will benefit from easier development of the lexical route to reading for words with irregular orthographies.

The results are certainly of potential interest, but I have a number of concerns at the design level, about the way the data are interpreted and about the data presentation, which I found to be muddled. Two issues in particular stand out:

1) The conception of the different types of noise employed in the study

2) Frequent unguarded statements that imply direction of causation, that cannot be determined from the data, in part because of an absence of relevant behavioural data

I shall outline the concerns I have below.

Fundamental Issues

I felt that there was at least one potentially fundamental flaw relating to the stimuli. The design of the experiment is articulated as a 2*2 parametric manipulation of informational * energetic masking. Unfortunately, it is simply not credible that masking can be non-energetic. Informational masking necessarily implies some degree of spectro-temporal coincidence of the target and the masker, in addition to the categorical coincidence (both are of the same nature as the target). A 'masker' that does not spectro-temporally overlap with a signal is not a masker, though it may have a deleterious impact on target processing for various reasons. The non-energetic non-informational condition is an interesting one, but since by design it aims to avoid masking the target it is difficult to see how these 4 'masking' conditions truly relate to one-another. It may be that the so-called non-energetic conditions are in fact distractors, but it is rather hard to determine what the conditions actually are based on the description in the methods in the main text. The supplementary materials only describe the generation of the energetic non-informational mask. More information is needed, and the terminology must be reconsidered. This has a considerable impact on the interpretation of the results. Furthermore, the difference between the masker types is not properly explored - to infer that it is the speech content of the informational conditions, and not their spectro-temporal modulation properties is not easily justified. This is, of course, a classical problem in the literature. How well are the non-informational and informational conditions really matched? Would a time-reversed version of the babble provoke the same kinds of effects, or not? It may be, of course that the authors do not want to draw specific conclusions based upon types of maskers. I would suggest that an effort be made to ensure that the rationale behind the masker design and their relationship to the hypotheses be made clearer. 

A basic theoretical issue that I found somewhat disconcerting is the underlying assumption of the desirability of higher cortical tracking. It is as yet unproven whether more tracking somehow equates to better performance. Despite the community's best efforts, the causal relationship between speech tracking and speech comprehension is at best unclear. 

Another serious concern is the complete absence of behavioural data on SiN. That is - without ascertaining that the differences in nCTS that are reported across groups and condition relate to SiN perception, claims cannot be made about the relationship between entrainment and SiN comprehension ability. This concern permeates the discussion. I do not question the various relationships that are reported between the nCTS and reading indicators but it does not seem tenable to make claims about how decreased AV fusion is responsible for impoverished SiN comprehension in dyslexic individuals. For instance: LL. 360-361 the measures are described as "objective cortical measure of the ability to deal with babble noise", this does not seem acceptable - where is the evidence that this reflects the ability to deal with babble noise? Do the participants report greater subjective clarity of the target, less effortful listening, higher accuracy of report? The one behavioural measure that seems to have been gathered is explicitly not analysed. I would strongly suggest that the authors reconsider this decision and attempt to link behaviour directly related to SiN to these measures in order to make their other conclusions more sustainable.

A general question concerns the decisions to use the lexical/phonological route distinction and to take the existence of separate routes as a given. It may be the case that this applies to alphabetic languages, but it is well known that it cannot generalise to non-alphabetic languages (e.g. Chinese). Some effort should be made to acknowledge that this study focuses on an alphabetic language with a non-transparent orthography, which may represent a specific subcategory of how reading can be implemented. 

The authors further analyse the difference between a dyslexic group two non-dyslexic groups: one a group of reading-level matched children and a second group of age-matched children. The outcomes of this analysis are not straightforward, but they are summarised as follows; dyslexic individuals show the same "reliance on visual speech to boost phrasal nCTS" as age-matched controls, but phrasal nCTS in babble and reliance on visual speech to boost syllabic nCTS are altered. These results could be made substantially clearer and conclusions can only meaningfully be drawn from them if an explicit comparison between the reading-matched controls and the age-matched controls is carried out, to determine whether these effects are in any way specific to dyslexia.

Major Issues

The repeated application of the PID methodology is challenging to follow. A little more effort to explain why each of the separate analyses is carried out, given that the method is presented as one that can provide insights into the unique contributions of a large set of variables. Naively, one would ask why all the variables of interest are not therefore handled together. Again, naively, one asks how legitimate it is to use LME to eliminate variables and then to use PID on selected variables only. To what extent could this be considered, in neuroimaging parlance, "double-dipping"? If the PID is intended only to provide qualitative insights then this is of no concern. It may be that it really is of no concern, and it would be welcome if the authors indicated this (and why).

There is a consistent lack of clarity in what the measures are, and this should be rectified. For example, Table 3 refers to the following: "Regressors included in the final linear mixed-effects model fit to the 5 reading scores". This is presumably not the correct description, since the factors listed in the table are the 5 reading scores. What is the dependent variable in this analysis?

Methodological queries

A lot of relevant information is assumed or relegated to the supplementary materials. It would be helpful to make some of the more crucial aspects of the methodology (e.g. stimulus generation, interpretation of PID) more obvious.

Numerous references are made to correcting variables for age, time in school, IQ, and then standardising (again). What was this correction? Would it not be possible to correct by the simple expedient of including the variables in the LME model?

Why did the model comparison procedure begin with the simplest rather than the maximal model?

Why was LME used for variable selection and not stepwise regression?

The statistics reported for the various PID analysis need further elucidation. It is not clear what the statistic is, what the degrees of freedom are, nor how the p values are derived. Consequently the p values seem inconsistent, e.g. LL 207-208: "(redundant information =0.16; p = 0.0020; synergistic information = 0.12; p = 0.26)", but in LL.206 unique information = 0.31 corresponds to p=.10. How are we to interpret these figures? I accept that this information may be in the various existing publications on the PID, but it would be extremely helpful to be able to interpret these values in context without referring to these.

It is not made clear why hemisphere is a variable of interest in the analyses - such a large-scale division of the brain seems somewhat arbitrary and should be clearly motivated.

Reviewer #2: This manuscript presents research aimed at investigating the links between reading ability and 1) the cortical tracking of speech (as measured using MEG) and 2) classic behavioural predictors of reading in a population of schoolchildren. The authors present children with stories that either have no noise added or four different types of noise and that either are accompanied by a relevant static picture or a video of the speaker's face. They then calculate a measure of how well the MEG is tracking the speech in each of the 8 noise conditions normalized by the tracking in the no noise condition. The authors also collected a large number of measures of reading performance and a large number of measures of classical behavioural predictors. They then use linear mixed-effect modelling to explore how any of their 8 cortical tracking measures - together with their many classical behavioural predictors - might explain reading performance. Furthermore, they use Partial Information Decomposition to identify whether any of these predictors makes a unique contribution to predicting reading performance or whether it might be redundant with other predictors or whether it might combine with other predictors to make even better predictions (synergy). They find a number of relationships between cortical tracking measures and behavioural predictors. And they show that some of these relationships (but not others) apply to individuals with dyslexia.

This manuscript tackles an interesting topic and does so with a nice data set and nice experiment. 

However, ultimately, I have one overarching concern that substantially dampens my enthusiasm for the work in its present form. Specifically, I could not help but worry about the robustness and replicability of the array of results we are presented with. The authors focus most of their analysis on 76 subjects. But they have 8 cortical tracking measures x 5 reading performance measures x 10 behavioral predictors (according to Table 1, but maybe only 5 in their analysis?). And, as such, I just found myself being sceptical about the results I was reading in every section. I would suggest that the authors might want to consider adding some additional analyses to reassure sceptical readers like me that the results we are seeing are likely to replicate. For example, the authors might consider permuting the labels on some of their predictors (e.g., the cortical tracking ones) and showing us that they can no longer get unique predictions from those cortical tracking measures. Or the authors might consider dividing their data in half and showing us that they consistently get the same pattern of results in both halves. 

Some more specific comments:

1) I think the nCTS equation should be included in the main body of the text.

2) Sorry if I missed it, but I did not see the authors discuss the fact that cortical tracking of speech will be (uninterestingly) improved by the inclusion of a video of the speakers face because of the contribution of correlated activity from visual (i.e., occipital) sensors. 

3) In line with my overarching concern above - I just found it implausible that nCTS could uniquely predict reading abilities when the classical behavioural predictors could not.

4) When you mention that "Two limitations are discussed in Supplementary Discussion", I think you should mention that they refer to limitations on only have one SNR for the stimuli and on training MEG models across all conditions and testing on each condition. Otherwise a reader is left wondering about/searching for those limitations.

5) As I read the discussion - I could not help but wonder what the authors might expect to see in the cortical tracking of illiterate adults. Surely their cortex will reliably track speech in noise, no? Is there any literature to suggest that illiterate adults struggle more in challenging listening environments?

Reviewer #3: This study looked systematically at the association between cortical tracking of speech in noise and reading skill in children. The authors found that cortical tracking of the phrasal content of speech in noise is differentially related to lexical reading strategies as opposed to sublexical reading strategies. There was also evidence of differences in the cortical tracking of speech in noise of children with dyslexia, suggesting that they better integrate visual speech information to improve processing of phrasal level speech tracking, rather than syllable-level.

Major points:

This was a novel and interesting study with some clear findings and I appreciated the chance to review it. In the interest of transparency, while I have some expertise in neuroimaging, I do not have expertise in MEG specifically. However, I was able to follow the procedure and analysis and, to the best of my knowledge, the methodology appeared robust. There are some details that I am seeking clarification on in this review but, on the whole, it seems to me that enough detail is included to allow replication and scrutiny of methods. The sample size is good for a study of this nature. I have some concerns about how aspects of the data are interpreted but, in general, conclusions do not go too far beyond the findings and add value to the existing literature base in this area. The manuscript was very clear and well-written and it was a thought provoking study.

I have some concerns around the dyslexic sample, however, and I think that the manuscript needs to provide more detail about this subgroup and the analysis strategy taken. It is not clear anywhere that I could find how the dyslexic group were defined and recruited. Was it on the basis of existing diagnosis, or screening tests as part of the research project? How homogenous were the group in terms of their reading difficulties? This is particularly important because inferences are drawn in relation to reading strategies using findings from the dyslexic group. I'm also unclear why the authors choose not to look at the relationships between CTSiN and reading skill in the children with dyslexia. I appreciate that, due to statistical power issues, they may not be able to conduct the same analysis as for the control group. However, in order to support some of the key interpretations of what CTS deficits in the dyslexic group mean that are proposed in the discussion, some idea of whether the relationships (even in terms of basic correlations) look similar seems vital to me. It would be hard to argue that CTS deficits are of importance in the dyslexic profile if they don't seem to relate to reading skill in this group. Similarly, can the authors provide details about the individual differences in CTS for the dyslexic group, as they do for the controls i.e. what percentage show statistically significant phrasal and syllable CTS? Important to know this is a reliable effect in that group in order to interpret their data.

A more minor point, but one that I think permeates several findings and discussions within the manuscript, is around the role of phonological awareness and how it has been tested. The relationships between reading and the phonological awareness measures are quite weak in this dataset. The authors rightly propose that this may be due to the age of the children and that phonological awareness becomes less central as reading becomes more automated. However, it is important for the authors to acknowledge the ceiling effects in their phonological awareness tests (~90% accuracy in control children, if I've interpreted tables correctly). It is much less likely that you will find phonological awareness mediates CTS effects if the tests are not sensitive enough, rather than because that skill does not mediate the relationship. I think that it is important that this is acknowledged as a possible reason why phonological awareness does not explain much of the variance in reading and why there may be no mediation effects. I think the conclusions relating to phonological processing need significant tempering because of this. In case of interest to the authors in their future work, we've found tests of spoonerisms to be more sensitive to phonological processing in these slightly older children who tend to perform towards ceiling on phoneme deletion or fusion tasks. 

Minor points

Line 41 - I think the authors should be cautious about claiming phrasal content of SiN relates to 'development of' lexical strategy when this is a concurrent association, not a longitudinal one. It is a little misleading.

Line 201 - is PID analysis robust to the fact that one set of variables has 5 indicators and the other has 8? Seems like this could bias the analysis, but I'm not particularly familiar with this analysis approach, so would appreciate the authors' clarification on this.

Line 220 - what does 'and further standardised' mean?

Section starting with line 215 - Table of correlations show that visual modulation of syllable nCTS very consistently correlated with reading measures. Why doesn't this come out in the linear mixed-effects modelling? Is it because it doesn't contribute anything unique? It would be helpful for this to come through more clearly somewhere in this section.

Lines 343-345 - The first and third relations referred to here seem to be to do with the link between CTS and reading skill so doesn't seem accurate to say that these were altered in the dyslexic group and relationships with reading skill weren't investigated in this group.

Lines 368-369 - I'm not clear how the results in dyslexia support this relation, particularly as children with dyslexia often have more significant difficulties with pseudoword reading than irregular word reading. Can this be clarified?

Line 518 - I think it's important to state the age range of the children somewhere here. I know it's in Table 1 but it's important information that needs to be found easily.

Line 708 - A (very brief) description of what nCTS is would be beneficial here. I know it's described in the results but despite the ordering of the sections many people will read the method before the results.

---

## [Decision Letter · Decision Letter 2]

7 Jul 2020

Dear Dr Destoky,

Thank you for submitting your revised Research Article entitled "Cortical tracking of speech in noise accounts for reading strategies in children" for publication in PLOS Biology. We've now obtained advice from one of the original reviewers and have discussed their comments with the Academic Editor. 

Based on the review, we will probably accept this manuscript for publication, assuming that you will modify the manuscript to address the remaining points raised by reviewer #2. Please also make sure to address the data and other policy-related requests noted at the end of this email.

We expect to receive your revised manuscript within two weeks. Your revisions should address the specific points made by each reviewer. In addition to the remaining revisions and before we will be able to formally accept your manuscript and consider it "in press", we also need to ensure that your article conforms to our guidelines. A member of our team will be in touch shortly with a set of requests. As we can't proceed until these requirements are met, your swift response will help prevent delays to publication.

*Copyediting*

*Published Peer Review History*

*Early Version*

*Submitting Your Revision*

Sincerely,

Roli Roberts

Roland G Roberts PhD

Senior Editor

PLOS Biology

on behalf of

Gabriel Gasque, Ph.D., 

Senior Editor

PLOS Biology

ETHICS STATEMENT:

-- Please include the full name of the IACUC/ethics committee that reviewed and approved the animal care and use protocol/permit/project license. Please also include an approval number.

-- Please include the specific national or international regulations/guidelines to which your animal care and use protocol adhered. Please note that institutional or accreditation organization guidelines (such as AAALAC) do not meet this requirement.

-- Please include information about the form of consent (written/oral) given for research involving human participants. All research involving human participants must have been approved by the authors' Institutional Review Board (IRB) or an equivalent committee, and all clinical investigation must have been conducted according to the principles expressed in the Declaration of Helsinki.

DATA POLICY:

We note that you plan to deposit your data in OSF; please could you deposit it and send us a reviewer link or password so that we can assess it. Note that as well as the raw data, we ask that all individual numerical values that underlie the data summarized in the figures and results of your paper be made available in one of the following forms:

Regardless of the method selected, please ensure that you provide the individual numerical values that underlie the summary data displayed in the following figure panels as they are essential for readers to assess your analysis and to reproduce it: Figs 2AB, 3, 4AB, 5, S2, S3AB. NOTE: the numerical data provided should include all replicates AND the way in which the plotted mean and errors were derived (it should not present only the mean/average values).

REVIEWER'S COMMENTS:

Reviewer #2:

Many thanks to the reviewers for their efforts in addressing my previous comments. I think the manuscript is significantly improved.

One remaining comment: I still remain skeptical about the idea of using 8 different cortical tracking measures and 5 different classical behavioral predictors of reading to account for 5 measures of reading in 73 subjects. And saying that you do so "in a single, statistically controlled analysis" doesn't ease my skepticism I am afraid. So I will leave it as a suggestion to the authors that they might want to consider how they can try to make their approach more compelling to new readers. This could be, for example, by discussing the strengths and weakness of PID for detecting spurious vs real relationships. Or by including some other analysis that would convince a new reader that the results are likely to replicate and are not simply the result of overfitting to the present dataset.

---

## [Editor Report · Decision Letter 3]

12 Aug 2020

Dear Dr Destoky,

On behalf of my colleagues and the Academic Editor, Timothy D. Griffiths, I am pleased to inform you that we will be delighted to publish your Research Article in PLOS Biology. 

Early Version

PRESS 

Kind regards,

Vita Usova

Publication Editor, 

PLOS Biology

on behalf of

Gabriel Gasque,

Senior Editor

PLOS Biology